# Lung microstructure in adolescent idiopathic scoliosis before and after posterior spinal fusion

Robert P. Thomen[1,2], Jason C. Woods[3,4], Peter F. Sturm[3,5], Viral Jain[3,5], Laura L. Walkup[4], Nara S. Higano[4], James D. Quirk[6,7], Brian M. Varisco[3,8]*

1 School of Medicine, University of Missouri, Columbia, Missouri, United States of America, 2 Division of Radiology, University of Missouri, Columbia, Missouri, United States of America, 3 College of Medicine, University of Cincinnati, Cincinnati, Ohio, United States of America, 4 Pulmonary Medicine, Cincinnati Children's Hospital, Cincinnati, Ohio, United States of America, 5 Orthopaedics, Cincinnati Children's Hospital, Cincinnati, Ohio, United States of America, 6 Mallincrodt Institute of Radiology, Washington University, St. Louis, MO, United States of America, 7 School of Medicine, Washington University, St. Louis, MO, United States of America, 8 Critical Care Medicine, Cincinnati Children's Hospital, Cincinnati, Ohio, United States of America

* brian.varisco@cchmc.org

**Data Availability Statement:** https://doi.org/10.6084/m9.figshare.12656723.

**Funding:** BMV: Scoliosis Research Society Investigatory Grant & Research Grant BMV:

## Abstract

Adolescent idiopathic scoliosis (AIS) is associated with decreased respiratory quality of life and impaired diaphragm function. Recent hyperpolarized helium (HHe) MRI studies show alveolarization continues throughout adolescence, and mechanical forces are known to impact alveolarization. We therefore hypothesized that patients with AIS would have alterations in alveolar size, alveolar number, or alveolar septal dimensions compared to adolescents without AIS, and that posterior spinal fusion (PSF) might reverse these differences. We conducted a prospective observational trial using HHe MRI to test for changes in alveolar microstructure in control and AIS subjects at baseline and one year. After obtaining written informed consent from subjects' legal guardians and assent from the subjects, we performed HHe and proton MRI in 14 AIS and 16 control subjects aged 8–21 years. The mean age of control subjects (12.9 years) was significantly less than AIS (14.9 years, p = 0.003). At baseline, there were no significant differences in alveolar size, number, or alveolar duct morphometry between AIS and control subjects or between the concave (compressed) and convex (expanded) lungs of AIS subjects. At one year after PSF AIS subjects had an increase in alveolar density in the formerly convex lung (p = 0.05), likely reflecting a change in thoracic anatomy, but there were no other significant changes in lung microstructure. Modeling of alveolar size over time demonstrated similar rates of alveolar growth in control and AIS subjects in both right and left lungs pre- and post-PSF. Although this study suffered from poor age-matching, we found no evidence that AIS or PSF impacts lung microstructure.

**Trial registration**: Clinical trial registration number NCT03539770.

Pediatric Orthopedic Society of North America/
NuVasive Spine Research Grant BMV: NIH/NHLBI
R01HL141229 BMV: NIH/NHLBI K08HL131261.

**Competing interests:** The authors have declared
that no competing interests exist.

## Introduction

Adolescent idiopathic scoliosis (AIS) affects 2–3% of American adolescents, and approximately ten percent of these children will require posterior spinal fusion (PSF) [1]. There are conflicting studies as to whether PSF improves pulmonary function [2–4], and pulmonary MRI studies demonstrate increased lung dimensions and improved diaphragm function after PSF for AIS [5]. However, whether any improvements are due solely to improved pulmonary mechanics or whether AIS and PSF might impact alveolar size, number, or structure has not been investigated. The lung contains progressively dividing airway structures that lead to gas-exchanging respiratory bronchioles and terminate in clusters of alveoli termed the acinus. The adult human lung has a gas-exchange surface area of 24 to 69 m2 [6] with 274–790 million alveoli [7]. The majority of alveologenesis occurs postnatally, and it is controversial whether this processes ceases during childhood [8] or continues throughout adolescence [9]—the age at which spinal curvature in AIS increases.

Mechanical forces influence distal lung structure during the alveolar stage of lung development. Rodents [10], large animals [11], and humans [12] that undergo pneumonectomy experience a compensatory increase in the size and number of alveoli of the remaining lung, and in mice, ipsilateral phrenic nerve transection after pneumonectomy prevents compensatory lung growth [13]. A rabbit model of early onset scoliosis demonstrated impaired lung microvascular development and emphysema in the concave lung [14]. Studies in infantile-onset scoliosis demonstrated impaired alveolar development, impaired gas exchange and differential lung vascularization [15, 16]. We developed this study to investigate whether altered intrathoracic forces in AIS (e.g. impaired diaphragm function, compression of the concave lung, and expansion of the convex lung) cause impaired alveolarization as has been noted in earlier-onset scoliosis. Convex and concave refers to lateral spine geometry when observed in the anterior-posterior or posterior-anterior plane.

Hyperpolarized gas MRI is a well-established method for evaluation of distal lung architecture [17]. In this technique the nuclear particle spins of a noble gas (3He in this case) are aligned using an optical spin transfer device yielding a nuclear magnetization more than 100,000-fold greater than could achieved by magnetic alignment. The hyperpolarized helium (HHe) is inhaled and the airspaces imaged using a standard multi-slice diffusion MRI sequence during a ~15 second inspiratory breath hold. The intrinsic diffusion of the gas within the lung is restricted by the alveolar walls, and diffusion imaging provides maps of apparent diffusion coefficients (ADC), alveolar number density (N), mean linear intercept ($L_m$), and other morphometric properties of the distal airspaces [18, 19]. This technique has been used to assess lung structure in normal children [9], children with bronchopulmonary dysplasia [20, 21], and adults with emphysema [22–24]. To determine whether patients with AIS experience alterations in distal lung structure and whether PSF reverses these changes, we performed HHe and proton MRI in control subjects and in AIS subjects recommended for PSF with follow up imaging at 1 year.

## Materials and methods

### Human subjects

The Cincinnati Children's Hospital Institutional Review Board (Approval 2013–8260) approved study procedures. Clinical trial registration number NCT03539770.

### Screening

Patients with AIS being evaluated for PSF in the orthopedics clinic at Cincinnati Children's Hospital were screened and approached for consent. Patients aged 8–21 years with AIS

were considered eligible with a Cobb angle of >50° and recommendation for PSF. Cobb angles were measured using standard PA spine radiographs and accepted techniques [25]. Exclusion criteria included previous spinal surgery, history of any chronic lung disease or asthma, personal history of smoking, supplemental oxygen requirement at baseline, born at <35 weeks gestational age, mechanical ventilation in the first year of life, or room air oxyhemoglobin saturation of less than 95%. Control subjects were recruited through community advertising with the same inclusion and exclusion criteria except that AIS was an exclusion.

## HHe MRI

All imaging was performed on a Philips Achieva 3.0T MRI with multinuclear capability. HHe was administered under FDA IND#122,670. 3He gas was polarized to approximately 50% polarization using a home-built rubidium optical spin transfer device [26]. Once polarized, HHe was diluted with nitrogen to a 50:50 volume ratio to ensure a consistent HHe intrinsic diffusion coefficient across subjects (D0 ≈ 1.2 cm2/s). The gas mixture was delivered to the subjects in the MR scanner and inhaled through a mouthpiece with two-way valve. Safety of this technique has been previously demonstrated [27, 28]. During the inspiratory breath-hold (FRC + 1 liter), 6–8 axial slices were acquired using a diffusion-weighted gradient echo MR sequence were acquired (b-values: 0, 2, 4, 6, 8 s/cm2, $\Delta = \delta$ = 1.5 ms, TR = 25 ms, TE = 4.9 ms matrix size = (31–54) phase encodes (PE) × (36–76) read-outs (RO)).

## Proton MRI

Ultrashort echo time (UTE) proton MRI images were obtained with multiple echo times (TE) using the 'stack-of-stars' radial acquisition scheme which was echo navigator gated at end expiration while breathing room air using the following parameters: repetition time (TR) = 5.8 ms, TE = 0.2/1.31/2.43 ms, flip angle (FA) = 5°, matrix size = 224×224, field of view (FOV) = 200×200 mm2, voxel size = 1.39×1.39×4 mm3 as previously reported [29].

## Image analysis

HHe images were analyzed according to previously described methods [17]. In short, each voxel's signal per b-value was fit to a multi-exponential model which provides7 quantitative, regional maps of HHe apparent diffusion coefficient (ADC [cm2/s]), alveolar number per unit volume (N [cm−3]), mean linear intercept (Lm [$\mu$m]), alveolar septal height (h, $\mu$m), and alveolar duct inner and outer radius (r and R respectively, $\mu$m). These maps were then segmented for separate left/right lung analysis. The UTE images with the longest echo-time (TE = 2.43 ms) were used to calculate right/left lung volumes since longer echo time images provide the greatest contrast between parenchyma and tissue. A threshold was applied to distinguish the lung interior from surrounding tissue, and the number of parenchymal voxels was counted for each lung; lung volume was calculated as the number of voxels multiplied by the voxel volume (7.73 mm3). For one case, UTE images were not acquired due to time restrictions. For this subject, lung volume was calculated using standard proton GRE sequences (voxel size = 1.95×1.95×10 mm3). Since lung volume increases with growth, we indexed lung volume measurements to the height-calculated functional residual capacity plus one liter [30].

$$FRC\ (mL) = 0.00175 * height(cm)^{2.66}$$

## Statistical analysis

The average value for all measurements from right, left, and bilateral lungs were used in analysis. Statistical comparisons were made using R version 6.2, ggplot2 and dply packages [31–33]. As many values were not parametric, we used Mann Whitney U-test to analyze all data with Wilcoxon Signed Rank test for sequential measurements in the same subject. For modeling of $L_m$ with age, the generalized linear modeling (glm) function of ggplot2 was utilized giving standard error ranges for modeled functions. p-values of less than 0.05 were considered significant.

## Results

### Demographics and enrollment

From April 1, 2015 to April 30, 2019 we enrolled 14 AIS and 16 control subjects. Data loss occurred in 1 AIS subject after the first scan and in 3 control subjects after the second scan. One AIS and 2 control subjects were lost to follow-up. One AIS subject had diffusion but not proton data available. Demographic data is show in Table 1.

### Imaging and data processing

A schematic depicting how the gradient echo HHe MR imaging quantifies airspace size is shown in Fig 1A and what derived measures represent in Fig 1B. Fig 1C shows maps of several different measurements from a single control subject's first imaging session.

### No difference in distal lung structure in AIS versus control

With the exception of expected anatomic differences between right and left lungs, proton and HHe image analysis at the first time point revealed no significant differences between control and AIS subjects (Table 2) or between left/concave or right/convex lungs (all AIS subjects had levoscoliosis Fig 2, Table 3) with regards to normalized lung volumes, mean linear intercept, alveolar density, septal height, inner or outer duct diameters, or surface to volume ratios.

### No meaningful changes in distal lung structure after PSF

Since alveolar size and number both increase during childhood [9], we evaluated how lung volumes, alveolar number, and microstructural measurements changed 1 year after PSF in AIS subjects and at one year after initial imaging in control subjects. In both control and AIS, there was an expected increase in lung volume and increase in alveolar size (Fig 3, Table 4). Post-PSF, reduced volume of the left/convex lung and increased alveolar density in the right/concave lung neared statistical significance when considered individually. In comparing the change in right vs. left lung measures, none were statistically significant though the relative lack of post-PSF convex lung growth is notable (Table 5).

**Table 1. Demographics.**

| | | Control (n = 16) | AIS (n = 14) | p-value |
|---|---|---|---|---|
| | Age (years) | 12.9 | 14.9 | 0.003 |
| | Sex (% Female) | 63 | 92 | 0.46 |
| | Height (cm) | 150.7 | 161.2 | 0.02 |
| Race | Caucasian (%) | 94 | 93 | 1 |
| | African American (%) | 0 | 7 | |
| | Asian (%) | 6 | 0 | |

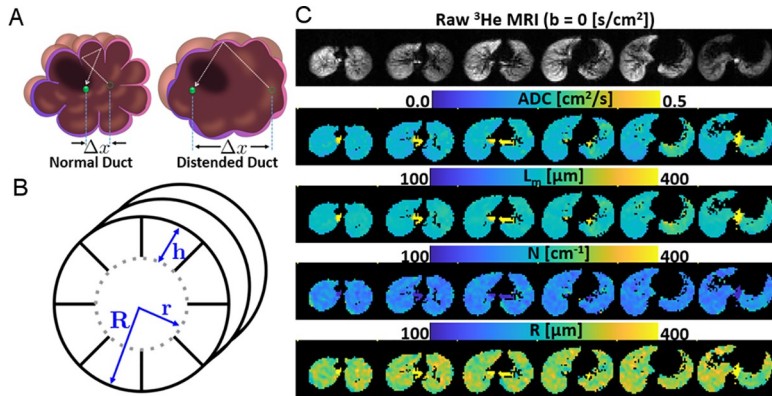

**Fig 1. HHe MRI overview.** (A) Schematic of a single HHe atom that is measured at to sequential time points. In a normal alveolar duct (top), although the atom diffuses the same total distance (dashed line) measurement is limited to the measured distance ($\Delta x$) and is relatively small. In dilated ducts, diffusion is less restricted and $\Delta x$ is larger. (B) Schematic representation of the outer alveolar duct radius (R), the inner duct radius (r), and the septal height (h). (C) Representative axial images of the raw HHe data (top), directly measured apparent diffusion coefficients (ADC), calculated mean linear intercept ($L_m$) values, derived alveolar density (N) values. Manual segmentation of right and left lung values was performed to obtain unilateral values for right-left comparisons.

## Discussion

In this small study of alveolar size, distal lung microstructure, and alveolar number in AIS, we found neither significant differences nor meaningful trends (a) between control and AIS subjects at baseline, (b) between concave and convex lungs of AIS subjects at baseline, or (c) in

**Table 2. Baseline differences between AIS and control.**

| | | Control (n = 16) | AIS (n = 13) | p-value |
|---|---|---|---|---|
| **Lung Volume (cm³)*** | **Right** | 735 | 833 | 0.53 |
| | **Left** | 609 | 702 | 0.86 |
| | **Bilateral** | 1350 | 1502 | 0.49 |
| **Normalized Lung Volume (to Predicted FRC+1L)#** | | 0.66 | 0.62 | 0.57 |
| **Lm (μm)*** | **Right** | 173.7 | 172.1 | 0.9 |
| | **Left** | 175.6 | 172.4 | 0.98 |
| | **Bilateral** | 171.6 | 172.9 | 0.82 |
| **Alveolar Number (10⁴)#** | **Right** | 12.34 | 12.07 | 0.74 |
| | **Left** | 9.66 | 10.09 | 0.9 |
| | **Bilateral** | 22.11 | 21.52 | 0.9 |
| **Alveolar Density (cm⁻³)*** | **Right** | 156 | 150 | 0.46 |
| | **Left** | 153 | 153 | 0.78 |
| **Septal Height (μm)*** | **Right** | 142.5 | 157.1 | 0.35 |
| | **Left** | 146.1 | 154.4 | 0.56 |
| **Inner Duct Diameter (μm)*** | **Right** | 138.8 | 134.4 | 0.9 |
| | **Left** | 132.9 | 135.2 | 0.9 |
| **Outer Duct Diameter (μm)*** | **Right** | 281.8 | 283.9 | 0.86 |
| | **Left** | 289.1 | 282.7 | 0.86 |
| **Surface to Volume Ratio (cm⁻¹)*** | **Right** | 232.4 | 233.9 | 0.86 |
| | **Left** | 238.3 | 233.3 | 0.86 |

*Directly measured values #Calculated values.

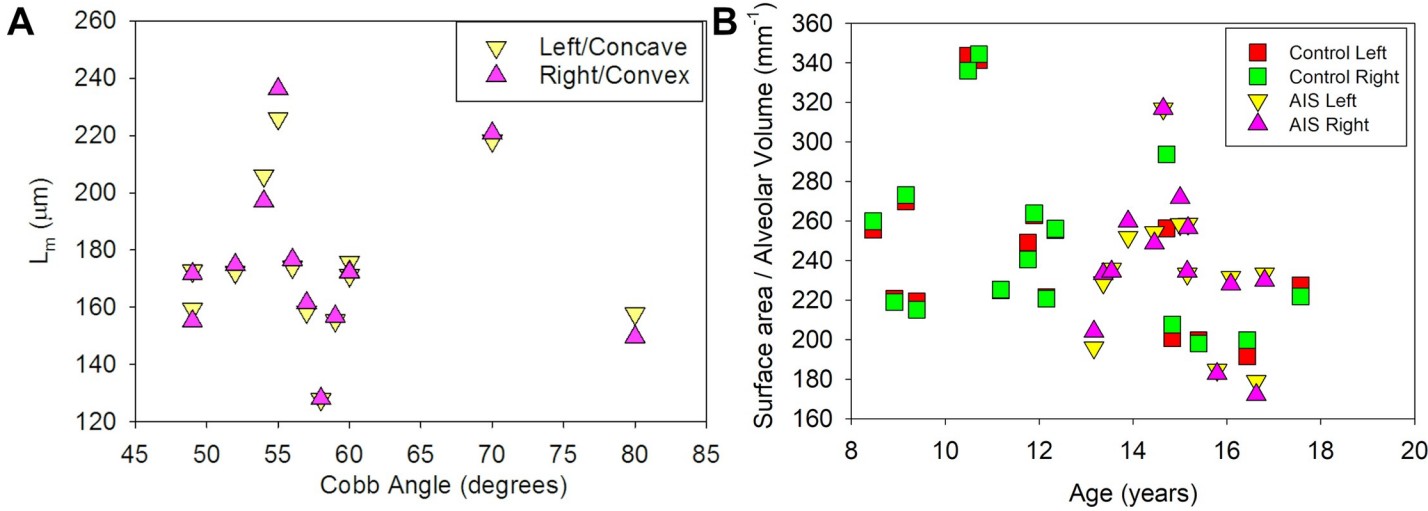

**Fig 2. Impact of AIS on distal lung architecture.** (A) All AIS subjects had levoscoliosis. The mean linear intercept ($L_m$) values were no different in left/concave vs right/convex lungs. (B) Alveolar surface to volume ratio decreased with age as expected since alveolar size increases and neoalveolarization slows during adolescence. There was no discernible difference between control and AIS groups or between right and left lungs with regard to this derived value.

change of alveolar microstructure at 1 year following PSF in AIS subjects. We found expected changes in lung volume and alveolar density in the formerly convex lung of AIS subjects following PSF. Our failure to find any microstructural changes in concave vs. convex lungs in AIS suggest that the previously reported changes in intrathoracic dynamics in AIS [5] do not impact alveolar development. Although the number of subjects is small, our findings do not support the hypothesis lung microstructure is changed in AIS.

Several study limitations should be noted. (a) Our use of a fixed inspired volume of HHe could have biased our findings to larger alveolar size in younger subjects. Using height and age normalized HHe volumes may have modestly reduced alveolar sizes in younger subjects, but this change would be predicted to reduce the already non-significant difference between AIS and control MLI. (b) The number of subjects enrolled was low and matching was poor. We had difficulty recruiting healthy females in the 15-18-year-old age range, and the age discrepancy between control and AIS groups make direct comparison difficult—particularly since increases in lung function measures are known to lag somatic growth in adolescence [34]. Subject matching and comparison of right and left lungs in the same subject somewhat alleviate

**Table 3. Right vs left lung comparisons in AIS vs. control.**

|  | Control (n = 16) | | | AIS (n = 13) | | |
|---|---|---|---|---|---|---|
|  | **Right** | **Left** | **p-value** | **Right** | **Left** | **p-value** |
| **Lung Volume (cm³)***| 735 | 609 | 0.08 | 833 | 702 | 0.05 |
| **Lm (μm)***| 173.7 | 175.6 | 0.83 | 173.1 | 172.4 | 1 |
| **Alveolar Number (10⁴)#** | 12.34 | 9.66 | 0.07 | 12.07 | 10.09 | 0.24 |
| **Alveolar Density (cm⁻³)***| 156 | 153 | 0.78 | 150 | 153 | 0.69 |
| **Septal Height (μm)***| 142.5 | 146.1 | 0.96 | 157.1 | 154.4 | 0.45 |
| **Inner Duct Diameter (μm)***| 138 | 132.9 | 0.96 | 134.4 | 135.2 | 1 |
| **Outer Duct Diameter (μm)***| 281.8 | 289.1 | 0.78 | 283.9 | 282.7 | 0.76 |
| **Surface to Volume Ratio (cm⁻¹)***| 232.4 | 238.3 | 0.93 | 233.9 | 233.3 | 1 |

*Directly measured values #Calculated values.

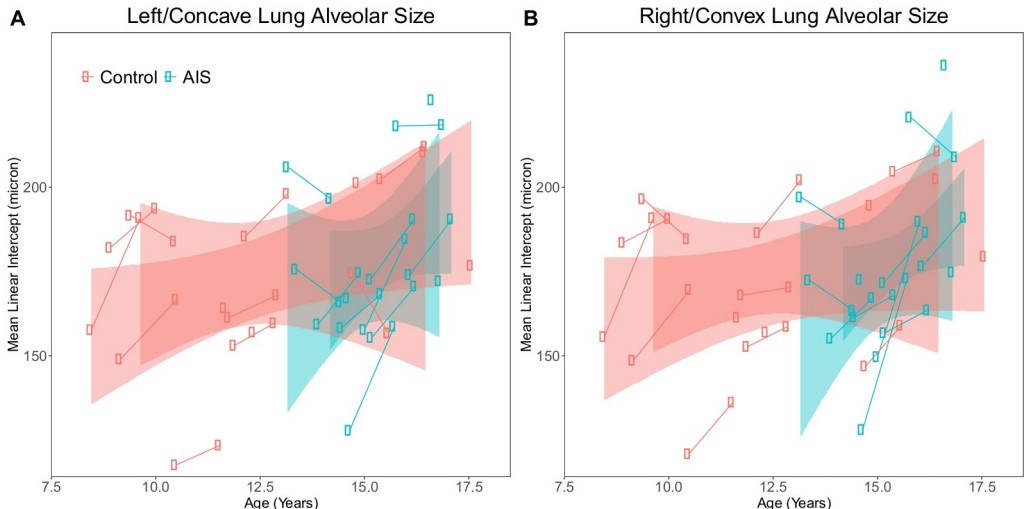

**Fig 3. Change in alveolar size at one year.** (A) In the left/concave and (B) right/convex lungs of control and AIS subjects, there was the expected increase in alveolar size over time without significant differences between them. Lines connect subjects and the shaded regions indicate standard error for both the first and 1-year HHe images as determined using generalized linear modeling.

the concern with poor age matching. (c) Follow-up time may have been insufficient. The one-year follow-up time was chosen based on normal post-operative clinic scheduling after PSF (the latest routine follow-up in the orthopedics clinic). It is possible that one year was insufficient follow-up to discover structural changes following PSF and to see whether the alveolar

**Table 4. One year post-PSF changes.**

|  |  | Control (n = 10) | AIS/PSF (n = 10) | p-value |
|---|---|---|---|---|
| Lung Volume (cm$^3$)* | Right | 45 | 46 | 0.24 |
|  | Left | 52 | 8 | 0.05 |
|  | Bilateral | 64 | 87 | 0.12 |
| Normalized Lung Volume (to Predicted FRC+1L)# |  | 0.07 | 0.11 | 0.88 |
| Lm ($\mu m$)* | Right | 9.6 | 6.7 | 0.34 |
|  | Left | 8.2 | 12.7 | 0.79 |
|  | Bilateral | 8.5 | 9.5 | 0.38 |
| Alveolar Number (10$^4$)# | Right | 0.05 | -0.04 | 0.96 |
|  | Left | 0.08 | -0.21 | 0.36 |
|  | Bilateral | 0.14 | -0.45 | 0.83 |
| Alveolar Density (cm$^{-3}$)* | Right | -22 | 2 | 0.05 |
|  | Left | -15 | -5 | 0.26 |
| Septal Height (μm)* | Right | -5.3 | -3.9 | 0.57 |
|  | Left | -8.8 | -3.7 | 0.17 |
| Inner Duct Diameter (μm)* | Right | 9.8 | 11 | 0.38 |
|  | Left | 9.2 | 16.7 | 0.46 |
| Outer Duct Diameter (μm)* | Right | 11.8 | 0.1 | 0.38 |
|  | Left | 8.9 | 4.3 | 0.79 |
| Surface to Volume Ratio (cm$^{-1}$)* | Right | -14.1 | -9.6 | 0.13 |
|  | Left | -9.9 | -17.4 | 0.27 |

*Directly measured values #Calculated values.

**Table 5. Change in right vs. left lung measurements in AIS vs. control.**

| | Control (n = 10) | | | AIS (n = 10) | | |
|---|---|---|---|---|---|---|
| | Right | Left | p-value | Right | Left | p-value |
| Lung Volume (cm³)* | 45 | 52 | 0.66 | 46 | 8 | 0.7 |
| Lm (μm)* | 9.6 | 8.2 | 0.91 | 6.7 | 12.7 | 0.91 |
| Alveolar Number (10⁴)# | 0.05 | 0.08 | 0.76 | -0.04 | -0.21 | 0.7 |
| Alveolar Density (cm⁻³)* | -22 | -15 | 0.08 | 2 | -5 | 0.62 |
| Septal Height (μm)* | -5.3 | -8.8 | 0.09 | -3.9 | -3.7 | 0.79 |
| Inner Duct Diameter (μm)* | 9.8 | 9.2 | 1 | 11 | 16.7 | 0.62 |
| Outer Duct Diameter (μm)* | 11.8 | 8.9 | 0.08 | 0.1 | 4.3 | 0.57 |
| Surface to Volume Ratio (cm⁻¹)* | -14.1 | -9.9 | 0.62 | -9.6 | -17.4 | 0.97 |

*Directly measured values #Calculated values.

density in post-PSF right/convex lung normalizes. (d) We did not evaluate the impact of kyphosis. (e) We did not perform pulmonary function testing because we felt that we were unlikely to see significant differences in this small cohort, and we did not wish to impose additional time burdens on families already committing two additional hours for our study after spending several hours in clinic. (e) We used non-parametric tests for some parametric data. Within most measures, some groups of data were parametric and others non-parametric. While applying a parametric test to some marginally significant results would have permitted crossing the significance threshold, we chose the above route for consistency and rigor.

## Conclusions

In summary, we found no evidence that AIS appreciably impacts alveolar size, alveolar duct morphometry, or alveolar number. The only changes that neared significance post-PSF were ones expected from spinal alignment.

## Acknowledgments

We thank Jennifer Anadio; Laura Benken; Rhonda Jones, RN; Kelli Krallman, RN; Lindsay Schultz, and Toni Yunger for their clinical and administrative support.

## Author Contributions

**Conceptualization:** Robert P. Thomen, Jason C. Woods, Peter F. Sturm, Viral Jain, Brian M. Varisco.

**Data curation:** Robert P. Thomen, Brian M. Varisco.

**Formal analysis:** Robert P. Thomen, Laura L. Walkup, Nara S. Higano, Brian M. Varisco.

**Funding acquisition:** Brian M. Varisco.

**Investigation:** Robert P. Thomen, Jason C. Woods, Peter F. Sturm, Viral Jain, Laura L. Walkup, Nara S. Higano, Brian M. Varisco.

**Methodology:** Robert P. Thomen, Jason C. Woods, Peter F. Sturm, Laura L. Walkup, Nara S. Higano, Brian M. Varisco.

**Project administration:** Jason C. Woods, Laura L. Walkup, Nara S. Higano, Brian M. Varisco.

**Resources:** Brian M. Varisco.

**Software:** Robert P. Thomen, James D. Quirk.

**Supervision:** Robert P. Thomen, Brian M. Varisco.

**Validation:** Robert P. Thomen, James D. Quirk, Brian M. Varisco.

**Visualization:** Robert P. Thomen, Brian M. Varisco.

**Writing – original draft:** Robert P. Thomen, Brian M. Varisco.

**Writing – review & editing:** Robert P. Thomen, Jason C. Woods, Peter F. Sturm, Viral Jain, Laura L. Walkup, Nara S. Higano, James D. Quirk, Brian M. Varisco.

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
