## [Decision Letter · Decision Letter 0]

12 Sep 2020

PONE-D-20-21934

Lung Microstructure in Adolescent Idiopathic Scoliosis Before and After Posterior Spinal Fusion

PLOS ONE

Dear Dr. Varisco,

Thank you for submitting your manuscript to PLOS ONE. After careful consideration, we feel that it has merit but does not fully meet PLOS ONE’s publication criteria as it currently stands. Therefore, we invite you to submit a revised version of the manuscript that addresses the points raised during the review process.

Both Reviewers mentioned the novelty of the study as well as the technical challenges that needed to be overcome in order to produce the dataset. Reviewer 1 had some concerns related to the presentation and interpretation of the results. Reviewer 2 had concerns related to methodology and power of the study and whether this has an impact conclusions of the study. Please respond to all of the critiques mentioned by both Reviewers.

We look forward to receiving your revised manuscript.

Kind regards,

Michael Koval

Academic Editor

PLOS ONE

Journal Requirements:

2. Please provide additional details regarding participant and parental consent. In the ethics statement in the Methods and online submission information, please ensure that you have specified whether consent was informed.

3. Please remove your figures from within your manuscript file, leaving only the individual TIFF/EPS image files, uploaded separately.  These will be automatically included in the reviewers’ PDF.

Reviewers' comments:

Reviewer's Responses to Questions

**Comments to the Author**

1. Is the manuscript technically sound, and do the data support the conclusions?

Reviewer #1: Yes

Reviewer #2: Partly

2. Has the statistical analysis been performed appropriately and rigorously? 

Reviewer #1: Yes

Reviewer #2: Yes

3. Have the authors made all data underlying the findings in their manuscript fully available?

Reviewer #1: Yes

Reviewer #2: Yes

4. Is the manuscript presented in an intelligible fashion and written in standard English?

Reviewer #1: Yes

Reviewer #2: Yes

5. Review Comments to the Author

Reviewer #1: Lung growth and adaptive change is an interesting topic and worthy of general readership.

Major comments

The premise of these experiments is a bit conflicted. The authors propose to investigate “whether any improvements (after surgery) are due solely to improved pulmonary mechanics or whether (surgery) impacts lung structure.” Further, the authors make a convincing argument for potential adaptive changes in alveolar structure. The experimental problem is that mechanical forces in the chest and alveolar microstructure are confounded in human studies. Since the alveoli are agnostic with respect to the spine--they only respond to intrathoracic forces--it is not at all surprising that the alveoli have adapted normally to their post-surgical mechanical environment. A positive result would have been surprising indeed! The authors should probably acknowledge this limitation in greater depth.

Minor comments

Table 1 is showing demographics. Why are p-values reported for some quantities in this table and not others? Presumably, they are trying to show to what extent the distribution of a particular parameter (i.e. age, sex, etc) is different between the two groups.

Not sure what they mean on page 9, line 118 where they differentiate between left/concave and right/convex lungs. My understanding is that both lungs have a convex (near the diaphragm) and concave (costal) surfaces and that these curvatures are not specific to a particular lung. I think they need to define what they mean by left/concave and right/convex.

Table 3: There seems to be an extra line in the table with “8” entered in one of the columns.

Figure 1. The last sentence in the figure caption has the text: “…and manual masking of right and left lungs (bottom).” I have no idea what to what the authors are referring here.

Figure 3. The label on this figure is misleading. It says “Change in Alveolar Size after PSF”. This can only be true for the AIS subjects undergoing a posterior spinal fusion (PSF) procedure. But they also show control values for the two scans separated by one year and certainly the controls did NOT undergo a PSF. Also, the figure inset shows the “control” data to be a black circle and the AIS data, a green circle. The figure only has pink and green symbols. Also, the shaded regions need to be defined as the uncertainty in the fit vs. age of the mean linear intercept. And they need to state what type of analytical expression was used to fit the data…was it a linear fit?

Reviewer #2: The authors set out to examine alveolar-airspace size in children with AIS, comparing them with healthy controls and to determine whether a specific surgical procedure (PSF) alters airspace characteristics.

The concept and methodology of the study is good. Authors make a good job of explaining complex concepts, and the various comparisons (control vs AIS baseline, left vs right, AIS pre and post PSF).

However, the various methodological flaws which have been acknowledged by the authors in the discussion prevent any meaningful conclusion.

1. The first major problem is the numbers of subject studied. It is acknowledged to be a ‘pilot’ study at various places. However, the authors attempt to draw conclusions in discussion section : e.g. ‘ While the study is not powered to prove the absence of an effect, the lack of any discernible trends in control vs AIS, concave vs. convex lung, or pre- and post-PSF imaging in AIS subjects argues against the presence of any such effect.’ (page 13, line 140-142). In my opinion (and from a statistical standpoint) there should not be any conclusion drawn from presence or absence of trends, where the differences do not reach a predefined (usually p<0.05) level of statistical significance.

2. The second problem is the difference in age and size of the control subjects (smaller and younger than AIS subjects.

a. This needs to be mentioned in the abstract section.

b. There has been an attempt to normalise using lung volumes derived from UTE proton MRI at TE = 2.43ms (page 7, line 79-81). If this technique has been described before (and compared to standard techniques), this should be referenced here.

c. Following this, normalisation has been attempted using NHANES III FVC values. I believe it is more appropriate to normalise by Functional residual capacity - FRC (or possibly FRC plus one liter, as this is the value at which measurements are taken)

d. Were any of the 3HeMRI derived parameters (Lm, alveolar number, etc.) normalised by volume? The values do not seem to be defined (don’t see any subscript for tables either).

3. The third problem is the relatively short interval between the primary and follow-up measurements. Authors do mention this in the discussion

a. Also, Adolescence is the period of growth spurt and rapid changes in height (and thoracic volume). It is well known that lung function (% predicted by height) lags behind physical growth. It is possible that alveolar dimensions also lag behind. This is especially important to consider because the control group are probably at a different stage of growth spurt compared to AIS group.

4. There is one conclusion that can be drawn from the fact that alveolar dimensions (alveolar density, septal height, ID diameter, OD diameter, S/V ratio) are similar between left and right lungs in the AIS group – that alveolar development should have continued after development of scoliosis (otherwise, the right side should have higher alveolar density).

6. PLOS authors have the option to publish the peer review history of their article (what does this mean?). If published, this will include your full peer review and any attached files.

Reviewer #1: No

Reviewer #2: No

---

## [Author Response · Author response to Decision Letter 0]

18 Sep 2020

September 18, 2020

To the Editor and Reviewers:

Thank you very much for your helpful critiques and suggestions on our manuscript “Lung Microstructure in Adolescent Idiopathic Scoliosis Before and After Posterior Spinal Fusion” (PONE-D-20-21934)

Below is a point-by-point response to critiques with reviewer text.

We hope that you are satisfied with these changes and look forward to a positive response.

Sincerely, Brian Varisco.

Associate Editor:

We have updated the format of the manuscript.

2. Please provide additional details regarding participant and parental consent. In the ethics statement in the Methods and online submission information, please ensure that you have specified whether consent was informed.

The informed consent and minors’ assent is now stated.

3. Please remove your figures from within your manuscript file, leaving only the individual TIFF/EPS image files, uploaded separately. These will be automatically included in the reviewers’ PDF.

These have been removed.

Reviewer #1: 

Lung growth and adaptive change is an interesting topic and worthy of general readership.

Major comments

The premise of these experiments is a bit conflicted. The authors propose to investigate “whether any improvements (after surgery) are due solely to improved pulmonary mechanics or whether (surgery) impacts lung structure.” Further, the authors make a convincing argument for potential adaptive changes in alveolar structure. The experimental problem is that mechanical forces in the chest and alveolar microstructure are confounded in human studies. Since the alveoli are agnostic with respect to the spine--they only respond to intrathoracic forces--it is not at all surprising that the alveoli have adapted normally to their post-surgical mechanical environment. A positive result would have been surprising indeed! The authors should probably acknowledge this limitation in greater depth.

It seems that we did not clearly articulate the linkage between spine curvature and intrathoracic forces that is the underlying premise of the study. The linkage between the spine curvature and pulmonary mechanics is based on a proton MRI study that found impaired diaphragm function in AIS and improvement post-PSF (Chu study). We made this clearer in the abstract and re-arranged and re-written the middle part of the introduction to make this premise more clear.

Minor comments

Table 1 is showing demographics. Why are p-values reported for some quantities in this table and not others? Presumably, they are trying to show to what extent the distribution of a particular parameter (i.e. age, sex, etc) is different between the two groups.

With only one non-white subject per group, we did not report the fact that differences in race between the groups were not statistically significant, but now we have done so.

Not sure what they mean on page 9, line 118 where they differentiate between left/concave and right/convex lungs. My understanding is that both lungs have a convex (near the diaphragm) and concave (costal) surfaces and that these curvatures are not specific to a particular lung. I think they need to define what they mean by left/concave and right/convex.

Thank you. Convex and concave lungs are orthopedic terminology with regards to spinal curvature. If the curvature. In the AP or PA plane, the lung that is being compressed is the convex lung and the lung that is being expanded is the concave lung. This has been clarified in the abstract and in the introduction.

Table 3: There seems to be an extra line in the table with “8” entered in one of the columns.

This was a formatting problem. All of the tables have been reformatted to fit journal specifications.

Figure 1. The last sentence in the figure caption has the text: “…and manual masking of right and left lungs (bottom).” I have no idea what to what the authors are referring here.

We have re-written this sentence to read that manual segmentation of right and left lung values was performed to obtain the values used in right-left comparisons.

Figure 3. The label on this figure is misleading. (a) It says “Change in Alveolar Size after PSF”. This can only be true for the AIS subjects undergoing a posterior spinal fusion (PSF) procedure. But they also show control values for the two scans separated by one year and certainly the controls did NOT undergo a PSF. (b) Also, the figure inset shows the “control” data to be a black circle and the AIS data, a green circle. The figure only has pink and green symbols. (c) Also, the shaded regions need to be defined as the uncertainty in the fit vs. age of the mean linear intercept. And they need to state what type of analytical expression was used to fit the data…was it a linear fit?

(a) The figure header was changed to read “Change in Alveolar Size at One Year.”

(b) With regard to the figure inset control data being a black circle and AIS data being a green (or aqua) circle), we think the reviewer is referring to the figure legend that is inset within panel A and that perhaps the black circle was a rendering issue. The downloaded Figure 3 TIF from the PLoS One submission portal has a dark pink square for control and an aqua square for AIS. When we double checked the submission PDF, the colors and shapes looked appropriate.

(c) We agree with the reviewer that an explicit statement that stating the use of generalized linear modeling is appropriate.

Reviewer #2: The authors set out to examine alveolar-airspace size in children with AIS, comparing them with healthy controls and to determine whether a specific surgical procedure (PSF) alters airspace characteristics.

The concept and methodology of the study is good. Authors make a good job of explaining complex concepts, and the various comparisons (control vs AIS baseline, left vs right, AIS pre and post PSF).

However, the various methodological flaws which have been acknowledged by the authors in the discussion prevent any meaningful conclusion.

1. The first major problem is the numbers of subject studied. It is acknowledged to be a ‘pilot’ study at various places. However, the authors attempt to draw conclusions in discussion section : e.g. ‘ While the study is not powered to prove the absence of an effect, the lack of any discernible trends in control vs AIS, concave vs. convex lung, or pre- and post-PSF imaging in AIS subjects argues against the presence of any such effect.’ (page 13, line 140-142). In my opinion (and from a statistical standpoint) there should not be any conclusion drawn from presence or absence of trends, where the differences do not reach a predefined (usually p<0.05) level of statistical significance.

As the reviewer likely appreciates, the study was accomplished stringing together several small grants and that without even the hint of a meaningful signal, our plans to leverage this preliminary data into a more definitive study were abandoned, but the data is still worth disseminating. However, the study wasn’t powered or designed to show absence of difference as the reviewer notes. We have modified the verbiage from “pilot” to “small” and modified our discussion to state that although limited by small numbers, our findings do not support the hypothesis AIS causes lung microstructural changes.

2. The second problem is the difference in age and size of the control subjects (smaller and younger than AIS subjects.

a. This needs to be mentioned in the abstract section.

We highlighted this shortcoming in the abstract

b. There has been an attempt to normalise using lung volumes derived from UTE proton MRI at TE = 2.43ms (page 7, line 79-81). If this technique has been described before (and compared to standard techniques), this should be referenced here.

This is the same technique that was reported in the Roach paper referenced in the sentence that the reviewer refers to. To make it more clear that this is the same validated technique that was previously reported, we moved the reference to the end and included verbiage to that effect.

c. Following this, normalisation has been attempted using NHANES III FVC values. I believe it is more appropriate to normalise by Functional residual capacity - FRC (or possibly FRC plus one liter, as this is the value at which measurements are taken)

This is a very good point and FRC plus one liter would be the ideal normalization value. That data is not available in NHANES, but a 1980 study using helium dilution provides FRC values for children (Buist, AJRCCM) and a 1990 study (Thorsteinsson, Anesthesiology) provide such values. We used the Thorsteinsson study as it contained a greater number of subjects and had tighter confidence intervals.

d. Were any of the 3HeMRI derived parameters (Lm, alveolar number, etc.) normalised by volume? The values do not seem to be defined (don’t see any subscript for tables either).

No, the raw measurements of these values were compared (for example cm^3 for lung volume, microns for Lm, cm^-1 for surface to volume ratio, all in parentheses). The reviewer’s point seems to be that it should be easier to look at the table and determine which parameters were directly measured vs. derived. We have added * and # with subscripts to make this more clear.

3. The third problem is the relatively short interval between the primary and follow-up measurements. Authors do mention this in the discussion

We agree that a 3-5 year follow-up would have been better, but since AIS patients are discharged from orthopedics clinic after 1 year follow-up, this was impracticable. 

a. Also, Adolescence is the period of growth spurt and rapid changes in height (and thoracic volume). It is well known that lung function (% predicted by height) lags behind physical growth. It is possible that alveolar dimensions also lag behind. This is especially important to consider because the control group are probably at a different stage of growth spurt compared to AIS group.

This is s a good point and statement to this effect has been added to discussion of weaknesses with a reference to Mahmoud et al 2018.

4. There is one conclusion that can be drawn from the fact that alveolar dimensions (alveolar density, septal height, ID diameter, OD diameter, S/V ratio) are similar between left and right lungs in the AIS group – that alveolar development should have continued after development of scoliosis (otherwise, the right side should have higher alveolar density).

This is true and has been more explicitly stated in discussion.

---

## [Editor Report · Decision Letter 1]

23 Sep 2020

Lung Microstructure in Adolescent Idiopathic Scoliosis Before and After Posterior Spinal Fusion

PONE-D-20-21934R1

Dear Dr. Varisco,

We’re pleased to inform you that your manuscript has been judged scientifically suitable for publication and will be formally accepted for publication once it meets all outstanding technical requirements.

Kind regards,

Michael Koval

Academic Editor

PLOS ONE

---

## [Editor Report · Acceptance letter]

29 Sep 2020

PONE-D-20-21934R1 

Lung microstructure in adolescent idiopathic scoliosis before and after posterior spinal fusion 

Dear Dr. Varisco:

I'm pleased to inform you that your manuscript has been deemed suitable for publication in PLOS ONE. Congratulations! Your manuscript is now with our production department. 

Kind regards, 

on behalf of

Dr. Michael Koval 

Academic Editor

PLOS ONE